# THE CONTEXT-AWARE LEARNER

## ABSTRACT

One important aspect of generalization in machine learning involves reasoning about previously seen data in new settings. Such reasoning requires learning disentangled representations of data which are interpretable in isolation, but can also be combined in a new, unseen scenario. To this end, we introduce the *context-aware learner*, a model based on the variational autoencoding framework, which can learn such representations across data sets exhibiting a number of distinct contexts. Moreover, it is successfully able to combine these representations to generate data not seen at training time. The model enjoys an exponential increase in representational ability for a linear increase in context count. We demonstrate that the theory readily extends to a meta-learning setting such as this, and describe a fully unsupervised model in complete generality. Finally, we validate our approach using an adaptation with weak supervision.

## 1 INTRODUCTION

Uncovering disentangled representations of data is a central goal of modern machine learning research. While practitioners are familiar with learning such representations, this comprises just one aspect of a two-sided problem; once we have successfully discovered meaningful concepts, the next step is to reassemble these in a novel and interesting manner. For example, humans are extremely adept at combining learned concepts in novel ways which may not have directly formed part of their experience. Naturally, achieving the second objective is contingent on discovering sufficiently powerful representations in the initial task.

As a thought experiment, imagine we have a collection of sets containing images of faces. Each set consists of faces which are consistent in some aspects e.g. one set features only male faces with blond hair, another features only female faces with glasses. This setting is analogous to an identity recognition task in a new context. Ideally, we would like to create disentangled representations of this data which reflect our high level impressions. The model should learn these in an unsupervised way. Moreover, we might hope that such a model would then be able to reason about joint distributions of which it has seen only some factor. For instance, having never been explicitly shown female faces with blond hair, but instead only examples of each features in isolation, it should be capable of generating suitable samples in the joint style. This is a novel unsupervised and data efficient regime, allowing for powerful modeling even when our data set fails to describe a distribution exhaustively.

The main contribution of this paper is the introduction of a model which can learn disentangled representations of distinct contexts exhibited across a number of data sets, and consequently reason about these representations in novel ways. We dub this model the context-aware learner. The advantage of our model is that it can not only learn representations of data by leveraging a latent variable framework, but the design lends itself naturally to combining these representations for interesting generalization capabilities.

## 2 PHRASING INFERENCE AS STOCHASTIC OPTIMIZATION

The structure of our model lies at the crossroads of stochastic variational inference and deep learning, and so we begin with an overview of work in these areas.

## 2.1 Stochastic Variational Inference and Deep Learning

Stochastic variational inference (Hoffman et al., 2013), a scalable method for performing inference large data sets, has received considerable attention in the deep learning literature of late. This is mainly due to the developments of amortizing inference networks and the reparameterization trick, introduced by Kingma & Welling (2013) and Rezende et al. (2014). The latter work coined the term stochastic backpropagation for propagating gradients through stochastic neural network nodes, while the former refers to the same idea as stochastic gradient variational Bayes (SGVB). While this method is often considered synonymous with the variational autoencoder (VAE) also introduced in Kingma & Welling (2013), we view the VAE as one particular instance of the more general class of models suggested by SGVB. In phrasing variational inference as a stochastic optimization problem, the above work allows us to bring the considerable machinery of deep learning to bear on the problem of inference in latent variable models.

## 2.2 The Variational Autoencoder

In the typical setup for stochastic variational inference using deep learning, we have a data set $\mathcal{X}$ consisting of observations $\mathbf{x}$. For each $\mathbf{x} \in \mathcal{X}$, the VAE assumes the existence of some latent variable $\mathbf{z}$ such that we may write the generative model

$$p_{\boldsymbol{\theta}}(\mathbf{x}) = \int p_{\boldsymbol{\theta}}(\mathbf{x}|\mathbf{z})p(\mathbf{z}) \, d\mathbf{z}, \tag{1}$$

where the likelihood is parameterized by $\boldsymbol{\theta}$. To infer $\mathbf{z}$, we introduce an encoder $q_{\boldsymbol{\phi}}(\mathbf{z}|\mathbf{x})$, which parameterizes an approximate posterior over the latent variable. We seek to learn the parameters $\phi$ and $\boldsymbol{\theta}$, and do so by bounding the single datum log marginal likelihood from below in the usual variational way (Jordan et al., 1999) i.e. find $\mathcal{L}_{\mathbf{x}}(\theta, \phi)$ such that

$$\log p_{\boldsymbol{\theta}}(\mathbf{x}) \geq \mathcal{L}_{\mathbf{x}}(\theta, \phi). \tag{2}$$

We then look to maximize $\mathcal{L}_{\mathcal{X}}(\theta, \phi) = \sum_{\mathbf{x} \in \mathcal{X}} \mathcal{L}_{\mathbf{x}}(\theta, \phi)$ across the entire training data set of examples, availing of the reparameterization trick to perform stochastic gradient descent.

## 2.3 The Neural Statistician

In the neural statistician (Edwards & Storkey, 2016), the above setting is tweaked for a more meta-take on representation learning. Now, we consider a collection of datasets $\mathcal{D}$. For each data set $\mathbf{D} \in \mathcal{D}$, we have observations $\mathbf{x}$, with a corresponding latent variable $\mathbf{z}$, as before. However, there is additionally a latent context variable, $\mathbf{c}$, which is held constant across items in each data set. That is, $\mathbf{c}$ is sampled once per data set, and $\mathbf{z}$ is sampled for each item. Assuming we have $N$ items per data set $\mathbf{D}$, the generative model can be written

$$p_{\boldsymbol{\theta}}(\mathbf{D}) = \int \left[ \prod_{n=1}^{N} \int p_{\boldsymbol{\theta}}(\mathbf{x}^{(n)}|\mathbf{z}^{(n)}, \mathbf{c})p_{\boldsymbol{\theta}}(\mathbf{z}^{(n)}|\mathbf{c}) \, d\mathbf{z}^{(n)} \right] p(\mathbf{c}) \, d\mathbf{c}. \tag{3}$$

The setting now closely resembles a deep learning twist on the classic topic model (Blei et al., 2003). This approach allows us to perform representation learning for entire data sets, and enables reasoning between data sets in the latent context space. The flexibility of SGVB means that once we specify an approximate posterior and its corresponding factorization, and thus find a variational lower bound $\mathcal{L}_{\mathbf{D}}(\theta, \phi)$ on the log marginal likelihood for each data set $\mathbf{D}$, it is possible to train the entire model end-to-end in much the same way as the VAE.

## 3 The Context-Aware Learner

Following the neural statistician, it is natural to question whether this meta-learning take on SGVB can be further extended. In particular, we might seek a model which could be applied in a setting where datasets exhibit multiple contexts $\{\mathbf{c}_{(k)}\}_{k=1}^{K}$, rather than a single shared context. Each observed dataset $\mathbf{D}$ would then consist of some subset of these which are held constant across the entire set, while the rest are free to vary per instance $\mathbf{x}$. The key point is that the contexts which are constant change per data set. Once more, the learning objective remains the same; we look to optimize a lower bound on the log marginal likelihood across the entire collection of datasets in order to learn the parameters $\phi$ and $\boldsymbol{\theta}$ of the inference and generative networks, respectively.

### 3.1 THE MODEL

Writing down the generative model for the context-aware learner becomes slightly more involved due to the presence of a categorical $\mathbf{a}$ variable, which must be introduced to indicate which contexts are constant, and which are varying in a particular data set. We use the convention $\mathbf{a}_k = 1$ to indicate that the $k^{th}$ context is constant. We also define the constant and varying index sets

$$A_c = \{k : \mathbf{a}_k = 1\} \qquad \text{and} \qquad A_v = \{k : \mathbf{a}_k = 0\}, \tag{4}$$

respectively. The generative model is only specified once $\mathbf{a}$ has been sampled, and this point is key. That is, the process is now dynamic per dataset $\mathbf{D}$ in our collection, and the inner 'plate' in Figure 1 is effectively free to move. There is not a single distribution over constant contexts, nor is there a single distribution over varying contexts. In fact, we have a total of $K$ distributions for the constant contexts, and likewise for the varying contexts. Once sampled, $\mathbf{a}$ selects which of these should be used in the generative process for each dataset.

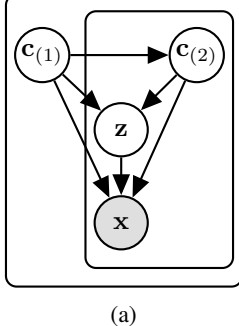 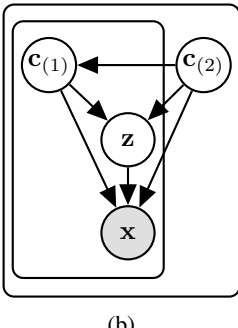

(a)           (b)

Figure 1: Graphical models describing two of four possible generative processes for the two-context case. Again, we emphasize that the generative model is not specified until $\mathbf{a}$ is sampled. (a) $\mathbf{c}_{(1)}$ is held constant while $\mathbf{c}_{(2)}$ varies. (b) Vice versa. Each edge in the model specifies a neural network, and multiple incoming edges concatenate their information for input to the receiving node. Each node outputs a parameterization of a distribution over its corresponding variable.

In the most general case, $\mathbf{a}$ is $aK$-dimensional binary vector, taking on $2^K$ possible values, corresponding to $2^K$ distinct generative models. In other words, it is equipped to handle datasets which have any subset of $K$ contexts held constant, including the extreme cases of all or none. There are natural simplifications to this setting, namely where we have a guarantee that some fixed number $K' \leq K$ of contexts are constant in each dataset, or simpler still, where we know exactly one context is held constant in each data set. We present the theory in full generality here.

We use the notation $\mathbf{c}_{(k)}^{(n)}$ to denote the $k^{th}$ context for the $n^{th}$ data point, and use $\mathbf{c}^{(n)}$ to refer to the complete vector of contexts for that data point, essentially grouping by omission of an index. We also denote the restriction to either the set of constant or varying contexts by writing $\mathbf{c}_{A_c}$, or $\mathbf{c}_{A_v}$, respectively.

The inclusion of a latent $\mathbf{z}$ variable is made so that the model has the capacity to represent aspects of the data which are neither strictly constant nor strictly varying. To encourage the variable to focus on this type of information, the distribution over $\mathbf{z}$ is not explicitly conditioned on $\mathbf{a}$. Instead, we condition solely on the context variables, so that $\mathbf{z}$ may learn useful features that are not represented by the contexts. In practice, we found that including $\mathbf{z}$ in the model resulted in a better variational lower bound, and more convincing samples. Moreover, we condition observations on both $\mathbf{c}$ and $\mathbf{z}$ so that $\mathbf{z}$ is free to absorb information not related to the context variables, an approach in the spirit of Maaløe et al. (2016).

The generative process for the context-aware learner is given by

$$p(\mathbf{D}) = \sum_{\mathbf{a}} p_{\boldsymbol{\theta}}(\mathbf{D}|\mathbf{a})p(\mathbf{a}) \tag{5}$$

where

$$p_{\boldsymbol{\theta}}(\mathbf{D}|\mathbf{a}) = \int \left[ \prod_{n=1}^{N} p_{\boldsymbol{\theta}}(\mathbf{x}^n|\mathbf{z}^{(n)}, \mathbf{c}^{(n)}) p_{\boldsymbol{\theta}}(\mathbf{z}^{(n)}|\mathbf{c}^{(n)}) \, \mathrm{d}\mathbf{z}^{(n)} \right] p_{\boldsymbol{\theta}}(\mathbf{c}|\mathbf{a}) \, \mathrm{d}\mathbf{c}. \tag{6}$$

Furthermore, we can factorize the context distributions using the Dirac delta as a point mass for the 'copying' of constant contexts, yielding

$$p_{\boldsymbol{\theta}}(\mathbf{c}^{(1)}|\mathbf{a}) = \prod_{k \in A_c} p_{\boldsymbol{\theta}}(\mathbf{c}_{(k)}^{(1)}|\mathbf{a}) \prod_{k \in A_v} p_{\boldsymbol{\theta}}(\mathbf{c}_{(k)}^{(1)}|\mathbf{c}_{A_c}, \mathbf{a}) \tag{7}$$

$$p_{\boldsymbol{\theta}}(\mathbf{c}^{(n)}|\mathbf{a}) = \prod_{k \in A_c} \delta\left( \mathbf{c}_{(k)}^{(n)} - \mathbf{c}_{(k)}^{(1)} \right) \prod_{k \in A_v} p_{\boldsymbol{\theta}}(\mathbf{c}_{(k)}^{(n)}|\mathbf{c}_{A_c}, \mathbf{a}), \; n > 1. \tag{8}$$

We make the assumption that the approximate posterior factors in an analogous way. It is important make the distinction here between the random variable $\mathbf{c} = \{\mathbf{c}_{(k)}\}_{k=1}^{K}$, and the choice we make to model this random variable with $2K$ distributions, $K$ of which handle the constant case, while the others are responsible for the varying case. In addition, we note that the varying context distributions are conditioned on the constant contexts, so that the model is better informed as to how exactly these varying aspects should be expressed.

With some work, it is possible to derive the variational lower bound for the single data set log marginal likelihood. We find it advantageous to decompose the bound into five parts, and moreover to separate the Kullback-Leibler (KL) divergence terms for the constant and varying contexts. It can be shown that

$$\mathcal{L}_{\mathbf{D}}(\boldsymbol{\theta}, \boldsymbol{\phi}) = R_{\mathbf{D}} - (L_{\mathbf{D}} + C_{v\mathbf{D}} + C_{c\mathbf{D}} + A_{\mathbf{D}}), \tag{9}$$

where

$$R_{\mathbf{D}} = \mathbb{E}_{\mathbf{z}^{(n)} \sim q_{\boldsymbol{\phi}}(\mathbf{z}^{(n)}|\mathbf{c}^{(n)}, \mathbf{x}^{(n)})} \left[ \mathbb{E}_{\mathbf{c}^{(n)} \sim q_{\boldsymbol{\phi}}(\mathbf{c}|\mathbf{a}, \mathbf{D})} \left[ \sum_{n=1}^{N} \log p_{\boldsymbol{\theta}}(\mathbf{x}^{(n)}|\mathbf{z}^{(n)}, \mathbf{c}^{(n)}) \right] \right] \tag{10}$$

$$L_{\mathbf{D}} = \mathbb{E}_{\mathbf{c}^{(n)} \sim q_{\boldsymbol{\phi}}(\mathbf{c}^{(n)}|\mathbf{a}, \mathbf{x}^{(n)})} \left[ \sum_{n=1}^{N} D_{KL}(q_{\boldsymbol{\phi}}(\mathbf{z}^{(n)}|\mathbf{c}^{(n)}, \mathbf{x}^{(n)}) \parallel p_{\boldsymbol{\theta}}(\mathbf{z}^{(n)}|\mathbf{c}^{(n)})) \right] \tag{11}$$

$$C_{v\mathbf{D}} = \mathbb{E}_{(\mathbf{c}_{A_c}, \mathbf{a}) \sim q_{\boldsymbol{\phi}}(\mathbf{c}, \mathbf{a}|\mathbf{D})} \left[ \sum_{n=1}^{N} \sum_{k \in A_v} D_{KL}(q_{\boldsymbol{\phi}}(\mathbf{c}_{(k)}^{(n)}|\mathbf{c}_{A_c}, \mathbf{a}, \mathbf{x}^{(n)}) \parallel p_{\boldsymbol{\theta}}(\mathbf{c}_{(k)}^{(n)}|\mathbf{c}_{A_c}, \mathbf{a})) \right] \tag{12}$$

$$C_{c\mathbf{D}} = \mathbb{E}_{\mathbf{a} \sim q_{\boldsymbol{\phi}}(\mathbf{a}|\mathbf{D})} \left[ \sum_{k \in A_c} D_{KL}(q_{\boldsymbol{\phi}}(\mathbf{c}_{(k)}^{(1)}|\mathbf{a}, \mathbf{D}) \parallel p_{\boldsymbol{\theta}}(\mathbf{c}_{(k)}^{(1)}|\mathbf{a})) \right] \tag{13}$$

$$A_{\mathbf{D}} = D_{KL}(q_{\boldsymbol{\phi}}(\mathbf{a}|\mathbf{D}) \parallel p(\mathbf{a})) \tag{14}$$

is the required lower bound. See Appendix A for details.

## 3.2 INCORPORATING WEAK SUPERVISION

Training the context-aware learner end-to-end hinges on a reliable method for integrating a categorical random variable into the SGVB framework. There are three aspects to this problem:

- successfully incorporating categorical reparameterization,
- determining the best method for collapsing across samples per data set in order to provide a good parameterization,
- overcoming the inherent symmetry in the assignment of contexts.

Categorical reparameterization has received recent attention for the training of VAE models with discrete latent variables (Rolfe, 2016). Both Maddison et al. (2016) and Jang et al. (2016) independently proposed the same relaxation of a categorical distribution. Tucker et al. (2017) recently built on these proposals, but none of these cases considered hierarchical models with several latent variables such

as the context-aware learner. In particular, the fact that the model may freely assign contexts in any order when learning $\mathbf{a}$ means that the solution is unique only up to permutation.

Though we have presented the theory in full generality above, one interesting narrowing of focus concerns the case where we know $\mathbf{a}$, a priori. This corresponds to a weak form of supervision in which we tell the model which data sets have a particular context held constant. The model must still learn representations which correspond to these factors of variation, but the task is now simpler since we do not need to worry about additionally inferring $\mathbf{a}$. Indeed, the $\mathbf{a}$ variable can be thought of a binary context label which informs the model of commonalities between data sets, but does not specify the exact nature of these common traits. The theory outlined above still applies; we just remove the KL term from the loss, and provide the model with a context label for each input. This aspect is the focus of our experiments, and allows for the interesting generalization capabilities we have outlined without needing to infer $\mathbf{a}$.

## 4 RELATED WORK

The context-aware learner touches on several areas of current research. Transfer learning remains a central interest in modern machine learning, and Pan & Yang (2010) provide a useful survey. Their parameter-transfer approach broaches similar topics to our work. We share distributions and network parameters across data sets, and look to leverage learned concepts to generate novel 'out-of-distribution' sets. A similar idea is presented in Bengio (2012), with a view to improving classification methods on test samples which may come from a different distribution.

Our model directly extends the ideas presented in the neural statistician (Edwards & Storkey, 2016), and consequently shares most in common with this approach. However, recent work has directly explored VAE extensions toward a similar goal. Bouchacourt et al. (2017) introduce the multi-level VAE, who seek to learn representations which disentangle factors of variation per groups with some common attribute. This is achieved through the introduction of style and content variables, which account for different aspects of the data. A closely related semi-supervised model is proposed by Siddharth et al. (2017), who also look to encode various facets of data into distinct latent variables. Our work differs in that it naturally extends to many factors of variation, or contexts.

Though we have not explored the generative adversarial network (GAN) (Goodfellow et al., 2014) in this work, it remains a highly influential and popular method for generative modeling. InfoGAN (Chen et al., 2016) takes an information theoretic interpretation of latent variables, and maximizes the mutual information between a conditioning variable and an observation. This results in a subset of the latent variables successfully disentangling factors of variation in an unsupervised fashion, which is useful when we do not possess common grouping knowledge as in the setting of the context-aware learner. Finally, Donahue et al. (2017) propose an algorithm which can learn subject identity, and a specific observation (e.g. configuration of lighting, pose etc.) of that subject, in a separate manner. Our model is capable of more granular representations, but again relies on meaningful grouping of the data in advance.

## 5 EXPERIMENTS

### 5.1 IMPLEMENTATION & ARCHITECTURE

All experiments were implemented using the PyTorch (Paszke et al.) framework, and computation was carried out on a single NVIDIA GeForce GTX 1070 GPU. We use the Adam optimizer (Kingma & Ba, 2014), batch normalization (Ioffe & Szegedy, 2015) on all layers, and the ReLU activation function. Training takes place over 300 epochs, with 10000 data sets sampled per epoch, a batch size of 32, and 10 samples per data set.

All approximate posterior and prior distributions over latent variables are diagonal Gaussian, each parameterized by a small fully connected network with skip connections between the first and last layers. To parameterize the constant contexts, we implement a 'statistic network' as in Edwards & Storkey (2016), which collapses across samples in a data set with a pooling operation.

The fact that the categorical variable $\mathbf{a}$ can take on $2^K$ values, and thus specify $2^K$ possible generative models, also introduces practical difficulties. A priori, we don't know which generative model we

need, even when we assume $\mathbf{a}$ is given to us. The solution we propose is to parameterize every distribution which may be used, and use $\mathbf{a}$ as a binary mask to zero out extraneous information. That is, we always generate a mean and log variance for each of the $K$ constant context distributions, and each of the $K$ varying context distributions, and use the mask implied by $\mathbf{a}$ to dynamically select the correct subset. In this way, the samples from the constant and varying context distributions complement one another; where one is zeroed out, the other has nonzero values, and vice versa. The mask also allows us to select the correct mean and log variance terms when computing the various KL divergence contributions.

Our implementation is bookended by a convolutional encoder and decoder, where the decoder structure mirrors that of the encoder. We use residual blocks (He et al., 2016), with specific choice of architecture similar to that outlined in Kingma et al. (2016). Downsampling is achieved by strided convolution, and we use bilinear upsampling as an inverse operation to mitigate possible checkerboard artifacts of transpose convolution. The encoder means that the model does not work with the data directly, but with extracted features. On the other hand, the decoder directly parameterizes $p_{\boldsymbol{\theta}}(\mathbf{x}^{(n)}|\mathbf{z}^{(n)}, \mathbf{c}^{(n)})$.

For the optimization criterion, we use a modified variational lower bound inspired by Sønderby et al. (2016), and we note that this was important for good performance. We begin training by inversely weighting the KL divergence terms, and increase the weighting over time so that we asymptotically maximize the true lower bound. This means that the approximate posterior distributions are initially more flexible, since it is less expensive for them to deviate from the prior. In particular, we use the criterion

$$\mathcal{L}_{\mathbf{D}}(\boldsymbol{\theta}, \boldsymbol{\phi}) = R_{\mathbf{D}} - \omega(L_{\mathbf{D}} + C_{v\mathbf{D}}) - \omega^2 C_{c\mathbf{D}}, \tag{15}$$

where $\omega = 1 + \alpha$, $\alpha = 1$ is the initial value, and $\alpha$ is annealed by a factor of a half after each epoch.

Full details of the implementation will be made available with a code release to accompany the paper. To maintain double-blind review, this is omitted while the paper is under consideration.

## 5.2 ROTATED MNIST

We first validate the proposed model using augmented MNIST (LeCun et al., 1998) data. We choose $K = 2$ contexts: digit and rotation. The digit can be any one of the ten MNIST classes, and we use four distinct rotations spaced evenly between $0°$ and $360°$. Both constant and varying contexts are 64-dimensional, while $\mathbf{z}$ is 16-dimensional. We use a learning rate of $4 \times 10^{-3}$, and a Bernoulli likelihood.

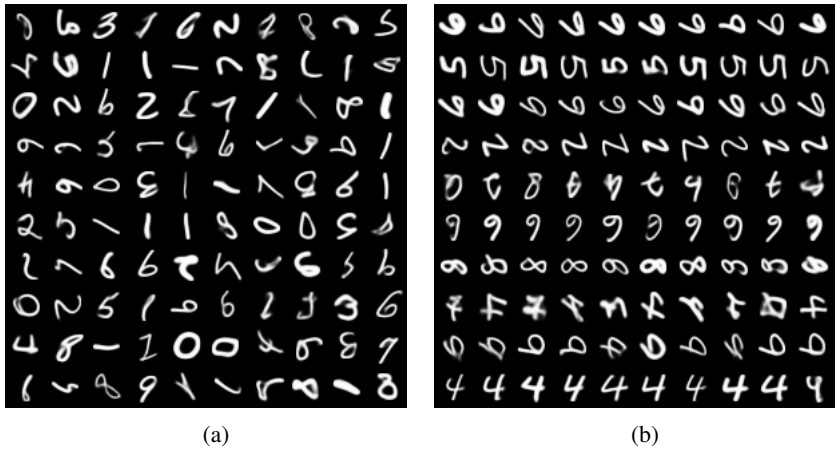

(a)  (b)

Figure 2: Generated samples from rotated MNIST. (a) The model is asked to generate data sets with neither contexts held constant, yielding sets of random digits with random rotations. (b) The model is asked to generate data sets with both digit and rotation held constant.

At test time, the design of the context-aware learner allows for interesting generalization capabilities. For instance, we might train the model using data sets that hold exactly one context constant, but at test

time ask it to produce samples of data sets with neither or both of these held constant. Furthermore, we might look to condition on two data sets with distinct contexts held constant, and synthesize a data set with the specific constant properties of both. These 'out-of-distribution' tasks are the main focus of our experiments.

Sampling new context combinations is straightforward; we provide the model with context labels it has not seen before, and examine the output of the generative process. Here, since each training data set has exactly one context held constant, corresponding to two-dimensional one-hot context labels, this means we are providing the model with either two-dimensional vectors of zeros, or two-dimensional vectors of ones. The results are shown in Figure 2. On the left, we have data sets consisting of varying digit and varying rotation. That is, there is no discernible pattern, since neither of the contexts are constant. On the other hand, the right hand side demonstrates samples which are consistent in both digit and rotation, and have both contexts held constant. We stress that the model has not been trained on either of these types of data sets, and can readily transfer the concepts it has learned to generate new combinations of contexts.

Figure 3: Conditional samples using rotated MNIST digits. The top two rows show the conditioning sets, while the bottom eight rows depict samples given these inputs. Note how the generated samples exhibit the constant aspects of both conditioning sets.

Synthesizing a data set from two others requires conditional sampling. In this case, we seek to calculate the approximate posteriors over constant contexts for the data sets with those corresponding contexts held constant. The mean of these distributions can then be passed through the generative process so that we condition on the most likely contexts given the data. The results are shown in Figure 3. The top row shows the data set with the constant digit '2', while the second displays the data set with constant rotation, $180°$. Thus, we would expect the model to produce data sets of the digit '2', rotated $180°$. Eight such synthesized data sets are shown in the remaining rows.

Table 1: Table showing classification accuracy (out of 100%) by digit and rotation pairs. We synthesize 250 samples of a given digit in a given rotation by generating data sets conditioned on each pair, as in Figure 3. The generated samples are then adjusted to the correct orientation and classified by a pre-trained network.

|  | 0 | 1 | 2 | 3 | 4 | 5 | 6 | 7 | 8 | 9 |
|---|---|---|---|---|---|---|---|---|---|---|
| $0°$ | 5.2 | 27.2 | 98.0 | 98.4 | 99.2 | 16.0 | 0.0 | 98.0 | 96.0 | 61.6 |
| $90°$ | 34.0 | 25.2 | 31.2 | 99.6 | 54.4 | 98.0 | 100.0 | 81.6 | 94.4 | 5.2 |
| $180°$ | 74.8 | 17.2 | 89.2 | 2.4 | 88.4 | 99.2 | 100.0 | 0.0 | 89.2 | 5.2 |
| $270°$ | 80.4 | 13.2 | 6.0 | 34.0 | 98.0 | 99.6 | 27.6 | 86.8 | 99.2 | 84.8 |

Due to the fact that in both the above cases the generated data sets were not contained in the training set, we can be sure the model is not simply memorizing training examples. However, we would still like to test the generated samples for authenticity. As such, we propose a final task as follows: if our model is generating digits with a particular rotation, then applying the inverse rotation to return

the digit to the correct orientation should yield examples which are classified correctly by some pre-trained MNIST classifier. Examining the the resulting label distribution can then be considered in the spirit of the Inception score pioneered by (Salimans et al., 2016).

With ten MNIST digits, and four rotations, there are forty total pairings possible. For each pair, we generate two data sets, constant in that digit and rotation, respectively. We condition on these to generate data sets consisting of that particular digit and rotation pair, yielding 250 total samples similar to Figure 3. Once we have correctly oriented these with the inverse rotation, we pass the samples through a simple four layer feed-forward convolutional network which achieves 99.02% test set accuracy on MNIST. The results over five runs are displayed in Table 1.

The distribution is such that most samples are amenable to correct classification, with some extreme outliers at the opposite end of the spectrum. For instance, we observed that in the case of generating the digit '3' upside down, the righted samples were overwhelmingly classified as the digit '8', leading to poor performance. There is also inherent confusion with inverting the digits '6' and '9', and we also noted that the corresponding samples for these classes were often confused by the model. Despite these difficulties, the results are promising, and suggestive that the model is indeed correctly identifying the disparate concepts of digit and rotation.

## 5.3 FACES

For our second task, we consider the CelebA (Liu et al., 2015) data set in order extend upon some aspects of the previous experiment. Using the annotations provided for CelebA, we select $K = 3$ contexts: smiling, wearing glasses, and wearing a hat. Both constant and varying contexts are 64-dimensional, while $\mathbf{z}$ is 16-dimensional, as before. We use a learning rate of $4 \times 10^{-4}$, and a Gaussian likelihood. The images are cropped to $64 \times 64$ in the standard way.

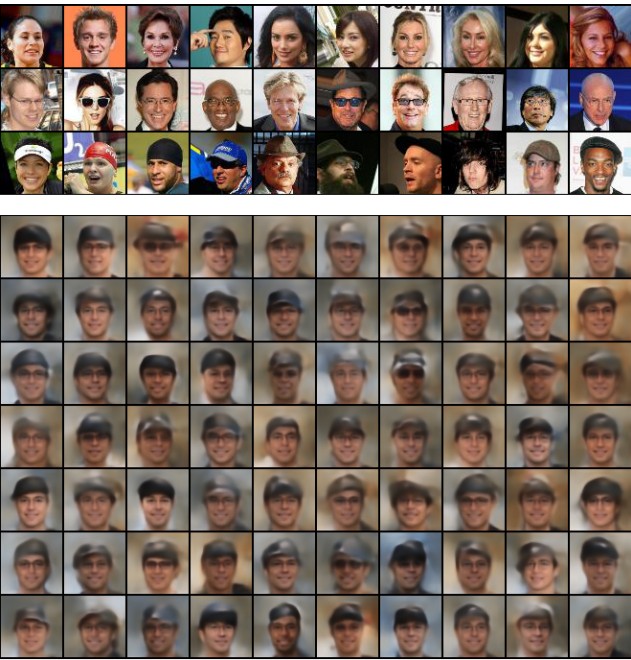

Figure 4: Conditional samples of faces from CelebA. The top three rows depict data sets which exhibit a shared context, namely smiling, wearing glasses, and wearing a hat, respectively. The bottom seven rows are generated by calculating the mean of the approximate posterior over the constant context for each conditioning data set, and sampling using these constant values. The goal is to generate data sets which are constant in all three contexts (i.e. smiling, wearing glasses, and wearing a hat). Note the model has only been trained on data sets with a single shared context, never multiple, and that we do not provide the model with information as to the nature of this context.

As before, we train the model using data sets with exactly one of these contexts held constant. Then, at test time, we examine the model's generalization abilities in the same manner, by asking for samples in the joint style. The results are shown in Figure 4.

Generally, we observe smiling faces, wearing both glasses and a hat. Although there is certainly room for improvement, we are not necessarily interested in generating high quality samples of faces. Though desirable, the main focus of the experiment concerns effective synthesis of new data sets which display the constant contexts exhibited in isolation by the training sets. In this respect, the results are encouraging. Once more, we emphasize that at training time, the model has not seen data sets with all three contexts held constant, but has nevertheless learned representations which facilitate generation from the joint distribution.

## 6 CONCLUSION

We have introduced the context-aware learner, a latent variable model suited to identifying patterns across data sets through the use of shared context variables. Our model naturally lends itself to a two-step interpretation: the initial learning of meaningful representations, followed by novel generative capabilities which utilize these learned concepts. We have outlined a general theory, and validated the proposed model on generalization tasks using both MNIST and CelebA.

ACKNOWLEDGMENTS

Omitted to maintain double-blind review.

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

## A   DERIVATION OF LOWER BOUND

Consider the KL divergence between the approximate and true posteriors over latent variables.

$$D_{KL}(q_\phi(\mathbf{z}, \mathbf{c}, \mathbf{a}|\mathbf{D}) \parallel p_\theta(\mathbf{z}, \mathbf{c}, \mathbf{a}|\mathbf{D})) = \sum_{\mathbf{a}} \int q_\phi(\mathbf{z}, \mathbf{c}, \mathbf{a}|\mathbf{D}) \log \frac{q_\phi(\mathbf{z}, \mathbf{c}, \mathbf{a}|\mathbf{D})}{p_\theta(\mathbf{z}, \mathbf{c}, \mathbf{a}|\mathbf{D})} \, \mathrm{d}\mathbf{z} \, \mathrm{d}\mathbf{c}.$$

We can use Bayes' rule to rewrite the true posterior in terms of the marginal, likelihood, and prior on latent variables, then use this to massage the KL divergence into a more amenable form.

$$D_{KL}(q_\phi(\mathbf{z}, \mathbf{c}, \mathbf{a}|\mathbf{D}) \parallel p_\theta(\mathbf{z}, \mathbf{c}, \mathbf{a}|\mathbf{D})) = \log p(\mathbf{D}) + D_{KL}(q_\phi(\mathbf{z}, \mathbf{c}, \mathbf{a}|\mathbf{D}) \parallel p_\theta(\mathbf{z}, \mathbf{c}, \mathbf{a}))$$
$$- \mathbb{E}_{(\mathbf{z},\mathbf{c}) \sim q_\phi(\mathbf{z},\mathbf{c}|\mathbf{D})} \left[\log p_\theta(\mathbf{D}|\mathbf{z}, \mathbf{c})\right].$$

The latter two terms in the expression on the right hand side are the KL divergence between the approximate posterior and prior over latent variables, and the reconstruction term in the form of an expectation over the log likelihood. It remains to write these terms in their most granular forms.

We begin with the likelihood. We have

$$\log p_\theta(\mathbf{D}|\mathbf{z}, \mathbf{c}) = \log \prod_{n=1}^{N} p_\theta(\mathbf{x}^{(n)}|\mathbf{z}^{(n)}, \mathbf{c}^{(n)})$$
$$= \sum_{n=1}^{N} \log p_\theta(\mathbf{x}^{(n)}|\mathbf{z}^{(n)}, \mathbf{c}^{(n)}),$$

and we can plug this directly into the expectation.

Next we decompose the KL divergence between the approximate posterior and prior. For the approximate posterior, we choose a factorization analogous to that specified for the generative model, meaning we are left with three KL divergence terms.

1. $D_{KL}(q_\phi(\mathbf{a}|\mathbf{D}) \parallel p(\mathbf{a}))$

2. $\mathbb{E}_{\mathbf{a} \sim q_\phi(\mathbf{a}|\mathbf{D})}\left[D_{KL}(q_\phi(\mathbf{c}|\mathbf{a}, \mathbf{D}) \parallel p_\theta(\mathbf{c}|\mathbf{a}))\right]$

3. $\mathbb{E}_{\mathbf{c} \sim q_\phi(\mathbf{c}|\mathbf{a},\mathbf{D})}\left[D_{KL}(q_\phi(\mathbf{z}|\mathbf{c}, \mathbf{D}) \parallel p_\theta(\mathbf{z}|\mathbf{c}))\right]$

We need only simplify terms 2 and 3. Beginning with term 2, we can split the context divergence into two parts, one corresponding to constant contexts, and the other to varying contexts.

$$D_{KL}(q_\phi(\mathbf{c}|\mathbf{a}, \mathbf{D}) \parallel p_\theta(\mathbf{c}|\mathbf{a})) = \sum_{k \in A_c} D_{KL}(q_\phi(\mathbf{c}_{(k)}|\mathbf{a}, \mathbf{D}) \parallel p_\theta(\mathbf{c}_{(k)}|\mathbf{a}))$$
$$+ \sum_{k \in A_v} D_{KL}(q_\phi(\mathbf{c}_{(k)}|\mathbf{c}_{A_c}, \mathbf{a}, \mathbf{D}) \parallel p_\theta(\mathbf{c}_{(k)}|\mathbf{c}_{A_c}, \mathbf{a}))$$
$$= \sum_{k \in A_c} D_{KL}(q_\phi(\mathbf{c}_{(k)}^{(1)}|\mathbf{a}, \mathbf{D}) \parallel p_\theta(\mathbf{c}_{(k)}^{(1)}|\mathbf{a}))$$
$$+ \sum_{n=1}^{N} \sum_{k \in A_v} D_{KL}(q_\phi(\mathbf{c}_{(k)}^{(n)}|\mathbf{c}_{A_c}, \mathbf{a}, \mathbf{x}^n) \parallel p_\theta(\mathbf{c}_{(k)}^{(n)}|\mathbf{c}_{A_c}, \mathbf{a})).$$

Here we have used the fact that for the constant context divergence terms, we can just take the sampled value for the first data point, since it is copied across the entire set.

Finally, we have

$$D_{KL}(q_\phi(\mathbf{z}|\mathbf{c}, \mathbf{D}) \parallel p_\theta(\mathbf{z}|\mathbf{c})) = \sum_{n=1}^{N} D_{KL}(q_\phi(\mathbf{z}^{(n)}|\mathbf{c}^{(n)}, \mathbf{x}^{(n)}) \parallel p_\theta(\mathbf{z}^{(n)}|\mathbf{c}^{(n)})).$$

Thus, the variational lower bound is given by

$$\mathcal{L}_\mathbf{D}(\boldsymbol{\theta}, \boldsymbol{\phi}) = R_\mathbf{D} - (L_\mathbf{D} + C_{v\mathbf{D}} + C_{c\mathbf{D}} + A_\mathbf{D}),$$

where

$$R_{\mathbf{D}} = \mathbb{E}_{\mathbf{z}^{(n)} \sim q_\phi(\mathbf{z}^{(n)}|\mathbf{c}^{(n)}, \mathbf{x}^{(n)})} \left[ \mathbb{E}_{\mathbf{c}^{(n)} \sim q_\phi(\mathbf{c}|\mathbf{a}, \mathbf{D})} \left[ \sum_{n=1}^{N} \log p_{\boldsymbol{\theta}}(\mathbf{x}^{(n)}|\mathbf{z}^{(n)}, \mathbf{c}^{(n)}) \right] \right]$$

$$L_{\mathbf{D}} = \mathbb{E}_{\mathbf{c}^{(n)} \sim q_\phi(\mathbf{c}^{(n)}|\mathbf{a}, \mathbf{x}^{(n)})} \left[ \sum_{n=1}^{N} D_{KL}(q_\phi(\mathbf{z}^{(n)}|\mathbf{c}^{(n)}, \mathbf{x}^{(n)}) \parallel p_{\boldsymbol{\theta}}(\mathbf{z}^{(n)}|\mathbf{c}^{(n)})) \right]$$

$$C_{v\mathbf{D}} = \mathbb{E}_{(\mathbf{c}_{A_c}, \mathbf{a}) \sim q_\phi(\mathbf{c}, \mathbf{a}|\mathbf{D})} \left[ \sum_{n=1}^{N} \sum_{k \in A_v} D_{KL}(q_\phi(\mathbf{c}_{(k)}^{(n)}|\mathbf{c}_{A_c}, \mathbf{a}, \mathbf{x}^{(n)}) \parallel p_{\boldsymbol{\theta}}(\mathbf{c}_{(k)}^{(n)}|\mathbf{c}_{A_c}, \mathbf{a})) \right]$$

$$C_{c\mathbf{D}} = \mathbb{E}_{\mathbf{a} \sim q_\phi(\mathbf{a}|\mathbf{D})} \left[ \sum_{k \in A_c} D_{KL}(q_\phi(\mathbf{c}_{(k)}^{(1)}|\mathbf{a}, \mathbf{D}) \parallel p_{\boldsymbol{\theta}}(\mathbf{c}_{(k)}^{(1)}|\mathbf{a})) \right]$$

$$A_{\mathbf{D}} = D_{KL}(q_\phi(\mathbf{a}|\mathbf{D}) \parallel p(\mathbf{a}))$$

