# OpenReview forum: "The Context-Aware Learner"
_ICLR.cc/2018/Conference — Reject_

### Official Review · AnonReviewer2 · 2017-11-27
**The authors propose an interesting disentangling version of the neural statistician with richer discrete structures.**

**Rating:** 6
**Confidence:** 5

**Review:**

The authors propose an extension to the Neural Statistician which can model contexts with multiple partially overlapping features. This model can explain datasets by taking into account covariate structure needed to explain away factors of variation and it can also share this structure partially between datasets.

A particularly interesting aspect of this model is the fact that it can learn these context c as features conditioned on meta-context a, which leads to a disentangled representation.
This is also not dissimilar to ideas used in 'Bayesian Representation Learning With Oracle Constraints' Karaletsos et al 2016 where similar contextual features c are learned to disentangle representations over observations and implicit supervision.

The authors provide a clean variational inference algorithm to learn their model. However, a key problem is the following: the nature of the discrete variables being used makes them hard to be inferred with variational inference. The authors mention categorical reparametrization as their trick of choice, but do not go into empirical details int heir experiments regarding the success of this approach. In fact, it would be interesting to study which level of these variables could be analytically collapsed (such as done in the Semi-Supervised learning work by Kingma et al 2014) and which ones can be sampled effectively using a form of reparametrization.

This also touches on the main criticism of the paper: While the model technically makes sense and is cleanly described and derived,  the empirical evaluation is on the weak side and the rich properties of the model are not really shown off. It would be interesting if the authors could consider adding a more illustrative experiment and some more empirical results regarding inference in this model and the marginal structures that can be learned with this model in controlled toy settings.
Can the model recover richer structure that was imposed during data generation? How limiting is the learning of a?
How does the likelihood of the model behave under the circumstances?
The experiments do not really convey how well this all will work in practice.

---

### Official Review · AnonReviewer3 · 2017-11-29
**A mixture of Neural Statisticians, lacks quantitative evaluation**

**Rating:** 4
**Confidence:** 3

**Review:**

This paper introduces a conditional variant of the model defined in the Neural Statistician (https://arxiv.org/abs/1606.02185). The generative model defines the process that produces the dataset. This model is first a mixture over contexts followed by i.i.d. generation of the dataset with possibly some unobserved random variable. This corresponds to a mixture of Neural Statisicians. The authors suggest that such a model could help with disentangling factors of variation in data. In the experiments they only consider training the model with the context selection variable and the data variables observed.

Unfortunately there is minimal quantitative evaluation (visualizing 264 MNIST samples is not enough). The only quantitative evaluation is in Table 1, and it seems the model is not able to generalize reliably to all rotations and all digits. Clearly, we can't expect perfect performance, but there are some troubling results: 5.2 accuracy on non-rotated 0s, 0.0 accuracy on non-rotated 6s. Every digit has at least one rotation that is not well classified, so this section could use more discussion and analysis. For example, how would this metric classify VAE samples with contexts corresponding only to digit type (no rotations)? How would this metric classify vanilla VAE samples that are hand labeled? Moreover, the context selection variable "a" should be considered part of the dataset, and as such the paper should report how "a" was selected.

This model is a relatively simple extension of the Neural Statistician, so the novelty of the idea is not enough to counterbalance the lack of quantitative evaluation. I do think the idea is well-motivated, and represents a promising way to incorporate prior knowledge of concepts into our training of VAEs. Still, the paper as it stands is not complete, and I encourage the authors to followup with more thorough quantitative empirical evaluations.

---

### Official Review · AnonReviewer4 · 2017-12-05
**interesting model presented, unclear paper**

**Rating:** 4
**Confidence:** 4

**Review:**

This paper proposes a model for learning to generate data conditional on attributes. Demonstrations show that the model is capable of learning to generate data with attribute combinations that were not present in conjunction at training time.

The model is interesting, and the results, while preliminary, suggest that the model is capable of making quite interesting generalizations (in particular, it can synthesize images that consist of settings of features that have not been seen before).

However, this paper is mercilessly difficult to read. The most serious problems are the extensive discussion of the fully unsupervised variant (rather than the semisupervised variant that is evaluated), poor use of examples when describing the model, nonstandard terminology (“concepts” and “context” are extremely vague terms that are not defined precisely) and discussions to vaguely related work that does not clarify but rather obscures what is going on in the paper.

For the evaluation, since this paper proposes a technique for learning a posterior recognition model, it would be extremely interesting to see if the model is capable of recognizing images appropriately that combine “contexts” that were not observed during training. The experiments show that the generation component is quite effective, but this is an obvious missing step.

Anyway, some other related work:
Lample et al. (2017 NIPS). Fader Networks. I realize this work is more ambitious since it seeks to be a fully generative model including of the contexts/attributes. But I mostly bring it up because it is an impressively clear presentation of a model and experimental set up.

---

### Decision · Program_Chairs · 2018-01-29
**ICLR 2018 Conference Acceptance Decision**

**Decision:**

Reject

**Comment:**

The paper proposes augmenting Neural Statistician with a meta-context variable that specifies the partitioning of the latent context into the per-dataset and per-datapoint dimensions. This idea makes a lot of sense but the reviewers found the experimental section clearly insufficient to demonstrate its effectiveness convincingly. Also introducing only the unsupervised version of the model, which looks challenging to train, but performing all the experiments with the less interesting semi-supervised version makes the paper both less compelling and harder to follow.